Millán-Callado

# Response characterization of Cosmic-Ray Neutron Sensors in neutron metrology reference fields

M. Ángeles Millán-Callado<sup>1</sup>, Roberto Méndez Villafañe<sup>2</sup>, Buthaina A. S. Adam<sup>1</sup>, Pavol Blahušiak<sup>3</sup>, Augusto Di Chicco<sup>1</sup>, Mirco Dietz<sup>1</sup>, Markus Köhli<sup>4,5</sup>, Benjamin Lutz<sup>1</sup>, Marcel Reginatto<sup>1</sup>, Zdenek Vykydal<sup>6</sup>, Jannis Weimar<sup>4,5</sup>, and Miroslav Zbořil<sup>1</sup>

**Correspondence:** M. Ángeles Millán-Callado (angeles.millan-callado@ptb.de)

**Abstract.** Soil moisture is recognized as an Essential Climate Variable (ECV) by the World Meteorological Organization (WMO), as it plays a key role in hydrology, agriculture, and climate by regulating land–atmosphere interactions such as evapotranspiration, groundwater recharge, and surface runoff. Despite its importance, existing monitoring methods operate at different spatial and temporal scales and remain poorly harmonized, leading to inconsistencies among datasets. The SoMMet (Soil Moisture Metrology) project addresses this gap by developing robust metrological tools to ensure consistent and comparable soil moisture observations.sensing detectors

Within this framework, the neutron response functions of commercial Cosmic Ray Neutron Sensing (CRNS) detectors (Hydroinnova CRS1000, Hydroinnova CRS2000/B, and StyX Neutronica S1) were characterized in radionuclide-based neutron reference fields ( $^{252}$ Cf, moderated  $^{252}$ Cf, and  $^{241}$ Am-Be), covering the full energy range relevant to CRNS applications. The experimental campaign was carried out at two neutron metrology laboratories: PTB (Physikalisch-Technische Bundesanstalt, Germany) and CIEMAT (Centro de Investigaciones Energéticas, Medioambientales y Tecnológicas, Spain). Monte Carlo simulations of the detector response were performed using MCNP and adjusted based on experimental data obtained at both facilities.

Independent validations were conducted at PTB and CIEMAT using monoenergetic neutron reference fields (from 24 keV to 5 MeV) and radionuclide reference sources, yielding agreement within 12 % between the experimental data and the validated simulations. These results provide the first benchmark data for CRNS detector models under reference conditions, representing a key step toward establishing robust SI-traceable calibration frameworks and harmonized deployment in soil moisture monitoring applications.

<sup>&</sup>lt;sup>1</sup>Physikalisch-Technische Bundesanstalt (PTB), Bundesallee 100, 38116 Braunschweig, Germany

<sup>&</sup>lt;sup>2</sup>Centro de Investigaciones Energéticas, Medioambientales y Tecnológicas (CIEMAT), Avda. Complutense 40, 28040 Madrid, Spain

<sup>&</sup>lt;sup>3</sup>Slovak Institute of Metrology (SMU), Karloveská 63, 842 55 Bratislava, Slovakia

<sup>&</sup>lt;sup>4</sup>Physikalisches Institut, Heidelberg University, Im Neuenheimer Feld 226, 69120 Heidelberg, Germany

<sup>&</sup>lt;sup>5</sup>StyX Neutronica GmbH, Cecil-Taylor-Ring 12-18, 68309 Mannheim, Germany

<sup>&</sup>lt;sup>6</sup>Czech Metrology Institute (CMI), Okružní 31, 63800 Brno, Czech Republic

#### 1 Introduction

Soil moisture strongly influences hydrological cycles, agricultural productivity, and land–atmosphere interactions. Its monitoring is therefore critical for climate change impact and adaptation strategies, weather predictions, and applications such as climate-smart agriculture, flood and drought forecasting, or water resources management (Vereecken et al., 2008). Recognizing this importance, the World Meteorological Organization (WMO) includes soil moisture among its Essential Climate Variables (ECVs) (WMO, UNEP, ISC, IOC-UNESCO, 2022).

Due to its important role, different methods have been developed for monitoring and measuring soil moisture at various spatial and temporal scales. Point-scale sensors are commonly used in agriculture and hydrology. These devices, installed at different depths, provide local information of water content that is not representative of large areas and therefore require dense networks or interpolation methods (Famiglietti et al., 2008), limiting their utility in atmospheric and land-surface applications. On the other hand, remote sensing methods, based on satellite products, can provide spatially extensive data (Manakos and Lavender, 2014), but only for surface moisture, and are affected by limitations such as coarse resolution in the kilometer range, lack of depth sensitivity, and disturbances due to vegetation, snow cover, or surface roughness (Wang and Qu, 2009).

In this context, intermediate-scale soil moisture measurement methods such as Cosmic-Ray Neutron Sensing (CRNS) (Zreda et al., 2008) provide an observational scale that complements ground-based sensors and satellite missions (Robinson et al., 2008), offering spatial footprints in the hectometer range while retaining sensitivity to subsurface soil moisture (Schrön et al., 2017). It is non-invasive and less sensitive to soil heterogeneity (Köhli et al., 2015), surface roughness, or vegetation (Desilets et al., 2010).

CRNS monitors soil moisture dynamics by detecting variations in the flux of epithermal albedo neutrons (from tens of meV up to about 1 MeV) resulting from the interaction of cosmic-ray-induced neutrons with the soil. These neutrons are moderated by hydrogen atoms in the soil, primarily from water molecules (Desilets et al., 2010), and their intensity is inversely correlated with soil moisture content (Köhli et al., 2021). CRNS detectors are designed to maximize efficiency in this energy range, typically incorporating large active volumes and hydrogenous moderators (Köhli et al., 2018), to increase the neutron detection efficiency for the comparably low environmental neutron flux.

CRNS devices are currently deployed in multiple monitoring networks, such as the COSMOS-UK (Bogena et al., 2022) and ADAPTER (Ney et al., 2021) networks. Despite the growing deployment of CRNS devices, systematic metrological standards and traceable calibration procedures remain underdeveloped. In particular, quantifying the impact of environmental conditions (e.g., soil type, humidity, temperature) on detector response is crucial to achieve reliable and comparable soil moisture estimates.

The present work is carried out within the framework of the European project SoMMet (Soil Moisture Metrology) (SOM, Visited on January 2025), funded under the European Partnership on Metrology of EURAMET. The initiative addresses the current lack of standardized approaches in soil moisture monitoring by providing a coordinated framework where different techniques and methods can be assessed, compared, and linked to metrological standards.

As part of the SoMMet project, a set of state-of-the-art commercial CRNS detectors has been experimentally characterized in neutron reference fields based on radionuclide sources (252Cf, moderated 252Cf, and 241Am-Be), covering the full energy range

relevant for CRNS. The experimental campaign was carried out at two neutron metrology laboratories: PTB (Physikalisch-Technische Bundesanstalt, Germany), a national metrology institute, and CIEMAT (Centro de Investigaciones Energéticas, Medioambientales y Tecnológicas, Spain), a designated institute for neutron measurements. The response functions of the detectors were obtained using Monte Carlo simulations with the MCNP code and corrected based on the experimental data acquired at both laboratories. In addition, an independent validation of the detector response, simulation model, and correction methods was performed using monoenergetic neutron reference fields at PTB, covering neutron energies from 24 keV to 5 MeV, traceable to primary neutron metrology standards (Nolte and Thomas, 2011a, b).

## 2 CRNS devices

65

75

CRNS devices used for soil moisture monitoring are based on conventional thermal neutron detectors, typically gaseous proportional counters (Kleinknecht, 1982) filled with  ${}^{3}$ He or  ${}^{10}$ BF $_{3}$  as the active medium (Knoll, 1979). In this experimental campaign, a boron-lined CRNS detector (Weimar et al., 2020) was also tested.

All of the devices investigated in this study are commercially available and incorporate integrated electronics and acquisition systems. Each provides a user interface that outputs the number of neutron counts over a selected time interval, along with relevant environmental parameters such as temperature, atmospheric pressure, and humidity, as well as diagnostic variables.

The list of devices tested during this experimental campaign is detailed below. Due to the commercial nature of these systems, additional details concerning internal geometry, data processing, or design cannot be disclosed:

- CRS1000 from Hydroinnova LLC (USA): Contains two <sup>3</sup>He detectors, one bare and one moderated using a high-density polyethylene (HDPE) casing, housed in a weatherproof metallic box that integrates all electronics and support systems.
   The external dimensions are 80 × 33 × 20 cm<sup>3</sup> (Fig. 7 (top)).
  - CRS2000/B from Hydroinnova LLC (USA): Comprises two <sup>10</sup>BF<sub>3</sub> detectors, one bare and one moderated with HDPE, each enclosed in separate weatherproof cylindrical housings. The detectors can be mounted together using a dedicated holder or operated independently. Each tube is 122 cm long and 14 cm in diameter (Fig. 3).
  - System S1 from StyX Neutronica GmbH (Germany): Comprises a single gas detector with a <sup>10</sup>B-coated conversion layer, moderated by HDPE, which serves as the detector housing. Additionally, it comes with a Gadolinium thermal neutron shield. The unit is 154 cm tall and 18 cm in diameter (Fig. 7 (bottom)).

All systems come with their own frontend and readout electronics, which for the StyX Neutronica system is documented (Köhli et al., 2024), for the CRS series see (Zreda et al., 2012).

The characterization measurements were performed at the neutron metrology laboratories of PTB and CIEMAT. The devices were distributed between the two facilities to allow for complementary measurements, taking advantage of their respective infrastructure and expertise.

The CRNS devices were irradiated in different orientations depending on the specific geometry of each unit. All measurements were performed using the standard configuration of the device, with the center of the irradiated surface aligned along the source-

to-detector axis using laser guides. Acquisition times were adjusted to ensure sufficient statistical significance in the neutron count rates.

Due to the low neutron flux in outdoor environments, CRNS detectors are typically large to maximize sensitivity. However, this size also limits the maximum count rate that can be processed without pile-up, saturation effects or electronic instabilities. In the case of the System S1, the original electronics saturated at around 20 cps for raw data transmission, an issue that was overcome by using an external multichannel analyzer (MCA). For the CRS1000 detector, where external electronic coupling was not possible, the counting rate was kept below 300 cps, a regime in which no significant loss of linearity was observed. In the case of the CRS2000/B, counting rates larger than 300 cps were measured, and possible signal losses were compensated via correction factors.

#### 95 3 Monte Carlo simulations

The three detectors studied, CRS1000, CRS2000/B and S1 have been simulated with MCNP6 (Version 6.1.0 (Pelowitz et al., 2013) for S1 and CRS1000, and version 6.2.0 (Werner et al., 2018) for the CRS2000/B). In each case, the neutron response for a specific energy range  $E_i$  is defined as:

$$R_{\rm MC}(E_{\rm j})[{\rm cm}^2] = \frac{M_{\rm MC}(E_{\rm j})}{\phi_{\rm MC}(E_{\rm i})},\tag{1}$$

where  $M_{MC}(E_j)$  is the number of simulated neutron reactions in the detector gas and  $\phi_{MC}(E_j)$  is the neutron fluence over the detectors's face of interest.

In all cases, the response function was simulated using a plane-parallel, monoenergetic neutron source incident perpendicularly on each face of interest, with energies ranging from the thermal neutron range up to  $100\,\text{MeV}$  and the number of reactions produced in each material has been simulated making use of the tally F4 and a multiplier FM to take into account the appropriate neutron reaction. ENDF/B-VIII.0 cross section library (Brown et al., 2018) and  $S(\alpha,\beta)$  treatment of low energy neutrons for polyethylene have been used. Each model is described in the next paragraphs. Due to manufacturer confidentiality, detailed specifications of the detector are not disclosed.

# 3.1 CRS1000 model

105

The CRS1000 consists of two  ${}^{3}$ He proportional counter detectors, each 2 inches (5.08 cm) in diameter and 33.9 cm in length: one moderated by 2.5 cm of high-density polyethylene (HDPE) and one left bare. In the simulations, the device was modeled as a single probe with two readouts. Neutron fluence was simulated on each external face of the aluminum housing  $(33 \times 19 \times 81.4 \text{ cm}^{3})$ , and the resulting count rates were calculated for each detector, without separating the contributions from

internal scattering. The output of each detector was defined as the number of neutron reactions occurring within the active volume. The geometry used in the simulations is shown in Fig. 1.

Figure 1. MCNP model of the CRS1000 device. The irradiation directions for the five relevant sides are indicated in the image at the right.

The response function was calculated for each face of interest: side 1 (towards the moderated detector), side 2 (towards the bare detector), the front (the largest surface, perpendicular to both detectors), and the top and bottom faces (parallel to both detectors and facing the smaller detector surfaces), as shown in Fig. 1. The results are shown in Fig. 2.

**Figure 2.** Example of the response functions calculated for the CRS1000 device for both internal detectors (moderated detector shown with solid lines, bare detector with dashed lines) and for all irradiation directions. These responses correspond to the MCNP model. The vertical dashed black lines indicate the monoenergetic energies used in this experimental campaign. The dotted black lines represent the response functions reported by Köhli et al. (2018), calculated using the URANOS code and scaled by an arbitrary factor.

The moderated detector exhibits a nearly flat response across the energy range of interest (1 eV to 1 MeV), whereas the bare detector is primarily sensitive to thermal neutrons. As expected, the detectors show anisotropic behavior, with significant differences in response depending on the irradiation direction. The shape of the simulated responses is consistent with previous data reported in the literature, as shown in Fig. 2, which includes the response curves published by Köhli et al. (2018) using the URANOS code (Köhli et al., 2023).

### 3.2 CRS2000/B model

The CRS2000/B system comprises two cylindrical detectors, as shown in Fig. 3, one of which is moderated by HDPE and the other is bare. In the definition of this device, we can distinguish between the tube, with an external length of 122 cm, a diameter of 14 cm; and the internal <sup>10</sup>BF<sub>3</sub> detector with a length of 85.4 cm and a diameter of 7.6 cm. HDPE surrounds this detector for the moderated tube and by air in the bare tube.

Figure 3. Picture of the CRS2000/B device and MCNP model. For both detectors, bare and moderated, only vertical and aligned directions are considered.

Each tube has been simulated in vacuum and irradiated only in two geometries, vertical and aligned, considering its cylindrical symmetry. In the vertical geometry, the direction of the incident neutrons is perpendicular to the tube axis, and in the aligned geometry, this direction is aligned with the tube axis. The response functions obtained for both detectors in the two geometries are shown in Fig. 3.

140

As in the previous model for CRS1000 the moderated detectors exhibit a nearly flat response across the energy range of interest, whereas the bare detectors are primarily sensitive to thermal neutrons for both geometries, vertical and aligned, being much more sensitive for the first one, where a larger cross-sectional area is shown to neutrons, than for the aligned geometry (see Fig. 4).

Figure 4. Simulated CRS2000/B bare and moderated detector response functions for both irradiation geometries.

# 3.3 System S1 model

The S1 detector is a boron-lined proportional counter of 125 cm long and 5.6 cm diameter embedded in a 2.5 cm thickness high-density polyethylene (HDPE) moderator to thermalize incoming neutrons. Additionally, the inner surface of the detector case is coated with a thin gadolinium oxide layer, which provides further shielding by capturing thermal neutrons and reducing the background from scattered low-energy neutrons. The external dimension of the device, 18 cm diameter and 154 cm length, is mostly due to the external moderator dimensions. The geometry implemented in the simulation is shown in Fig. 5.

Two irradiation geometries were considered: with the neutron beam perpendicular to the detector's axis (vertical configuration) and parallel to it (aligned configuration), as is shown in Fig. 6. Compared to the CRS1000 and CRS2000/B detectors (see Fig. 2 and 3), the S1 system shows a sharper decrease in the thermal energy region, due to the presence of an additional gadolinium (Gd<sub>2</sub>O<sub>3</sub>) layer.

Figure 5. MCNP model of the S1 system detector.

**Figure 6.** Simulated S1 detector response functions for both irradiation geometries, with the CRS1000 response under FRONT irradiation included for qualitative comparison.

## 4 Irradiations at the radionuclide reference fields: validation of the MC models

## 4.1 Experimental setup at the radionuclide reference fields at PTB

At PTB, calibration measurements were performed for the CRS1000 and System S1 devices using well-characterized radionuclide neutron sources:  $^{252}$ Cf(sf),  $^{252}$ Cf(sf) with a D<sub>2</sub>O moderator sphere of 15 cm radius and a 1 mm Cd filter ( $Q = 8.49 \cdot 10^4 \text{ s}^{-1}$ , same source with and without the moderator sphere), and  $^{241}$ Am-Be( $\alpha$ ,n) ( $Q = 2.921 \cdot 10^6 \text{ s}^{-1}$ ). These sources are installed in a dedicated irradiation bunker at PTB, with internal dimensions of 7 m × 7 m × 6.5 m (Kluge, 1998). The walls and ceiling are made of 1 m thick reinforced concrete. The irradiation geometry allows for remote source positioning and automated control of detector distances. During measurements, the sources were suspended near the geometrical center of the room.

**Figure 7.** CRS1000 (top) and System S1 (bottom) in the PTB irradiation room during irradiation with the moderated <sup>252</sup>Cf source. The moderator sphere, with the Cd foil facing the detector, is also visible in both images.

At the calibration positions, the relative standard uncertainty of the neutron fluence ranges from 1% to 3%, depending on the characteristics of the neutron field. It is derived from the calibrated source strengths and evaluated through neutron spectrometry and transport calculations, in accordance with ISO 8529-1 (International Organization for Standardization, 2021).

For the CRS1000, measurements were conducted in five geometrical orientations: front, side 1, side 2, top, and bottom (see Fig. 1). All irradiations were performed at a fixed source-to-detector distance of 170 cm, measured from the center of the neutron source to the center of the detector box.

For the System S1, irradiations were carried out in top and side orientations (with the detector either parallel or perpendicular to the source-to-detector axis, respectively) at three source-to-detector distances: 120 cm, 170 cm, and 220 cm, always referenced to the geometrical centers of the source and the detector.

In similar experimental setups, a shadowing object is typically used to block the detector and separate the background neutron component from the direct signal (Obeid et al., 2021). In the present measurements, no suitable object was available to fully shadow the detector. As a result, both direct and scattered neutron components contributed to the recorded signal. To account for these background effects, a detailed MCNP simulation was performed, incorporating the full detector geometry within a comprehensive model of the experimental room.

**Figure 8.** MCNP model of the PTB irradiation room showing the main structural components. Although only the case for the CRS1000 is shown in the picture, both detector geometries (see Fig. 1 and Fig. 5) were included in the simulations to replicate the experimental conditions.

This MCNP room model (Al Qaaod et al., 2024), previously validated at PTB using the NEMUS Bonner sphere spectrometer, includes multiple material layers in the walls, floor, and ceiling, as well as key structural elements such as the water tank beneath the floor (used for source storage), the room door, and detector support structures (Fig. 8).

180

190

# 4.2 Experimental setup at the radionuclide reference fields at CIEMAT

The irradiation room at CIEMAT is a bunker with dimensions of  $7.5 \,\mathrm{m} \times 9 \,\mathrm{m} \times 8 \,\mathrm{m}$  and concrete walls of  $1.25 \,\mathrm{m}$  thickness, with a 3 m long bench where the equipment to be calibrated is positioned (Guzman-Garcia et al., 2015). The neutron sources are stored under water, and an automated system allows their remote manipulation from the control room to position them in the calibration position corresponding to the geometrical center of the room. The laboratory has well-characterized radionuclide neutron sources currently used for calibration purposes:  $^{252}$ Cf,  $D_2$ O-moderated  $^{252}$ Cf without Cd,  $H_2$ O-moderated  $^{252}$ Cf, and  $^{241}$ Am-Be neutron sources, and other smaller  $^{241}$ Am-Be employed for verification.

In a first approach at the CIEMAT facility, the CRS2000/B system has been measured using this  $^{241}$ Am-Be source (labelled as AmBe (3)) and applying the shadow cone technique to determine direct contribution. The corrected emission rate on the measurement date is  $Q = 6.343 \cdot 10^5 \, \mathrm{s}^{-1}$  and corresponds to the weakest source in this installation. From this emission rate, the fluence rate at the irradiation position can be determined, and for a distance of 200 cm is  $1.35 \, \mathrm{cm}^{-2} \mathrm{s}^{-1}$ . Two shadow objects have been used, one conventional shadow cone for aligned irradiations, and the other with a pyramidal geometry to be applied for vertical irradiations, see Fig. 9.

Figure 9. Irradiation at CIEMAT of one of the CRS2000/B tubes with the shadow object with pyramidal geometry.

In order to complete the validation of the model, a series of simulations to determine the response of the CRS2000/B to this isotopic neutron source at CIEMAT has been performed. For this simulation, it is not necessary to include the facility, and basically it consists of a convolution of the response function with the ISO spectrum associated with the <sup>241</sup>Am-Be neutron source. These results can be compared with those obtained through measurements with and without shadow objects, but as in the irradiations at PTB with isotopic neutron sources, the results show that the scattered component is not properly discriminated with the available shadow objects, and a second approach has to be adopted, considering a detailed model of the laboratory.

In this case, we can compare the obtained results with the measurements without the shadow cone because the effect of walls, ground, roof, air, and other elements like the irradiation bench are already considered. So, the simulation corresponds to the direct + scatter case.

In this simulation, the neutron sources are described in detail, and the elements present in the laboratory, like the bench supports, the stairs, or the calibration table, are considered, as can be shown in Fig. 10. For the definition of the neutron source, a detailed model developed to evaluate its anisotropy has been included (Méndez Villafañe et al., 2019).

**Figure 10.** MC model of CIEMAT installation showing details of the bench, calibration table, source support and capsule holder, and sources storage pool.

## 4.3 Experimental results: adjustment of the MC models

To compare the experimental results with the simulation values, a dimensionless ratio  $S_i$  was defined for each irradiation configuration i:

$$S_i = \frac{C_i \cdot Q_i}{D_i}. (2)$$

Where  $C_i$  is the number of simulated neutron reactions in the detector gas per source neutron,  $Q_i$  [ $s^{-1}$ ] is the source emission rate of the source used in the corresponding irradiation, and  $D_i$  [ $s^{-1}$ ] is the experimental detection rate. The ratio was defined based on the source strength rather than the fluence over the detector, since the latter showed significant variations along the detector tubes due to the large size of the devices relative to the irradiation room.

The mean value  $\overline{S}$  of the  $S_i$  was determined for each device based on 30 reference irradiations for the CRS1000 (bare and moderated detector), 15 for the System S1, and 14 for the CRS2000/B. The standard deviation was used in all cases to estimate

230

the uncertainty, yielding:

210 
$$\overline{S}_{CRS1000} = 0.79 \pm 0.07$$
,

$$\overline{S}_{\text{CRS2000/B}} = 1.34 \pm 0.05,$$

$$\overline{S}_{S1} = 1.26 \pm 0.16$$
.

In the ideal case of full agreement between the experimental data and the simulations, this ratio would be equal to 1. Nevertheless, the average ratio  $\overline{S}$  obtained with MCNP across all sources and irradiation configurations shows systematic discrepancies. These are mainly attributed to limitations in the completeness of the model geometries. Furthermore, the magnitude of such deviations fluctuates depending on the neutron field under consideration, as the energy-dependent mismatch between the simulated and the actual detector response impacts each dataset differently. As a result, calibrations performed with multiple sources — despite leading to a larger apparent dispersion — provide a more realistic estimate of the underlying uncertainties than those based on a single source, which underestimate the variability by neglecting energy-range effects.

This emphasizes the need to cross-check and validate the digital models in well-known traceable neutron fields before using them to study the detector behavior in unknown conditions. The systematic deviation from the experimental data can be corrected by applying the average ratio  $\overline{S}$  as a global adjustment factor in the MC simulations. The simulation-to-measurement ratio for the different irradiation setups was redefined as follows:

$$S_i' = \frac{C_i/\overline{S} \cdot Q_i}{R_i} \tag{3}$$

The corrected  $S'_i$  values are shown in Fig. 11, with error bars representing one standard deviation (k = 1). The error bars include experimental uncertainties (less than 1%, accounting for statistical and source strength uncertainties), MC statistical uncertainty (less than 2%), and the uncertainty of the scaling factor  $\overline{S}$  due to the dispersion of the  $S_i$  values.

# 5 Irradiation at the monoenergetic reference fields at the PTB accelerator facility PIAF

The characterization of the models prepared independently by PTB and CIEMAT was carried out in two steps. First, the MC adjustment coefficient was determined using reference irradiation data, as described in previous sections (see section 4.3). In a second step, the adjusted model was used to calculate the response under different conditions, in order to validate that both the adjustment factor and the corrections applied to the experimental data remain valid beyond the specific calibration scenarios.

For this purpose, the detectors were irradiated at the Ion Accelerator Facility (PIAF) of PTB. Monoenergetic neutron reference fields were produced using ion beams delivered by the 2 MV tandetron accelerator operated in continuous (DC) mode, in compliance with ISO 8529 (International Organization for Standardization, 2021).

**Figure 11.** Simulation-to-measurement ratio for all irradiation configurations using the different radionuclide reference fields for the CRS1000 device (top), CRS2000/B (middle), and the System S1 (bottom).

255

Fields with nominal peak energies between 24 keV and 5 MeV were produced in a low-scatter experimental hall using the <sup>7</sup>Li(p,n), <sup>3</sup>H(p,n), and <sup>2</sup>H(d,n) reactions (Nolte et al., 2004; Nolte and Thomas, 2011a). These fields are generated in open geometry with well-defined source-detector distances. For this study, six neutron energies were selected, as summarized in Table 1.

| E <sub>ion</sub> /MeV | Reaction                          | Angle [°] | E <sub>n</sub> / MeV |
|-----------------------|-----------------------------------|-----------|----------------------|
| 1.941                 | $^{7}\text{Li}(p,n)^{7}\text{Be}$ | 80        | 0.024                |
| 1.941                 | $^{7}$ Li(p,n) $^{7}$ Be          | 0         | 0.144                |
| 2.021                 | $^{7}\text{Li}(p,n)^{7}\text{Be}$ | 0         | 0.250                |
| 2.297                 | $^{7}$ Li(p,n) $^{7}$ Be          | 0         | 0.565                |
| 2.049                 | $^{3}$ H(p,n) $^{3}$ He           | 0         | 1.2                  |
| 2.360                 | $^{2}$ H(d,n) $^{3}$ He           | 0         | 5.0                  |

**Table 1.** Monoenergetic neutron beams used in the characterization campaign.

A long-term stable <sup>3</sup>He-filled proportional counter, traceable to the PTB primary standard, was used as a transfer instrument to determine the absolute neutron fluence at the position of the instruments under test (Nolte and Lutz, 2024). The uncertainty in the neutron fluence was below 3 % for all fields except the 5 MeV field, for which it was 7 %.

To minimize background from room-scattered neutrons, the production target was positioned above a gridded floor with an open pit underneath, and surrounded by large distances to all structural surfaces (walls, floor, and ceiling). Nevertheless, given the intrinsic sensitivity of CRNS detectors to low-energy neutrons, the background contribution was determined and corrected using the shadow cone technique (Obeid et al., 2021; Paneru et al., 2023) (see Fig. 12).

The detectors were placed at 6 meters from the neutron production target at a neutron emission angle of  $0^{\circ}$ , except for the 24 keV energy, which, dictated by the reaction kinematics, was measured at  $80^{\circ}$ . In this case, the detector had to be positioned at 4 meters due to geometrical constraints of the experimental setup. Due to the asymmetry in the detector response, irradiations were carried out in the same orientations as those defined in previous sections (see Fig. 1).

According to the simulations (see Fig. 2), the bare detectors are considerably more sensitive — by more than an order of magnitude — to the low-energy background from scattered neutrons in the room than to the direct beam. Based on shadow cone measurements, the background contributes on average 90 % of the counting rate in bare detectors for irradiations on the front and side faces, and more than 95 % (up to 99 % depending on the energy) for the top and bottom faces. Combined with the low statistics arising from the low efficiency of the bare detector at these energies, this leads to large uncertainties and discrepancies between measured and expected values. Therefore, for the irradiations in the monoenergetic neutron beams, useful experimental data could only be obtained for the moderated detectors.

For the S1 detector, due to the inherently low detection efficiency observed in the aligned configuration (see Fig. 6), only data from the vertical configuration (beam perpendicular to the tube axis, i.e., detector in working position) were used for validation.

According to the shadow cone method, the inclusion of the Gd neutron absorber layer in this system reduces the background contribution to approximately 20 %, compared to the 50–70 % observed in the other devices.

**Figure 12.** Low-background experimental hall of PIAF during one of the irradiations. The CRNS device and the shadow cone in a typical experimental configuration are also shown.

# 5.1 Experimental response and validation of the MC models

The experimental results, obtained from irradiations with the monoenergetic neutron beams at PIAF and the calibration sources at CIEMAT (all reference neutron fields), are used to validate the calibration of the digital model of each detector. This ensures that the adjustment factors and corrections applied are reliable not only under the specific calibration conditions but also when extrapolated to other scenarios, thus establishing a validated framework for future studies to assess the influence of environmental factors under outdoor conditions (e.g., buildings, water bodies, snow cover) on the signal response.

Analogously to the definition of the simulated detector response in equation 1, the experimental detector response,  $R_{\rm exp}$ , is calculated from the experimental data as:

$$270 \quad R_{\rm exp}[{\rm cm}^2] = \frac{M_{\rm i}}{\phi_{\rm i}}, \tag{4} \label{eq:exp_exp}$$

where  $M_i$  is the instrument output (counts) and  $\phi_i$  is the neutron fluence over the detector [cm<sup>-2</sup>] for an specific configuration.

To compare experimental results with the adjusted simulations, a dimensionless ratio  $\overline{T}$  between simulated and experimental responses is defined:

$$\overline{T} = \frac{R_{\rm MC}/\overline{S}}{R_{\rm exp}}.$$
 (5)

Obtaining, for each device:

$$\overline{T}_{CRS1000} = 1.00 \pm 0.11,$$

$$\overline{T}_{CRS2000/B} = 1.02 \pm 0.11,$$

285

$$\overline{T}_{S1} = 1.06 \pm 0.17.$$

The average ratios for each device are consistent with unity for all energies and irradiation configurations, within the experimental uncertainties, indicating that the model adjustment is accurate and that the calibration procedure is validated for the tested conditions.

Fig. 13 shows the ratio  $\overline{T}$  for all the validation irradiation configurations performed on the detectors, with an average agreement within 10%. The largest fluctuations and uncertainties are observed at the top and bottom faces, where detector efficiency is lower and the fraction of counts from the direct beam is reduced, leading to larger statistical variations. In these orientations, the MCNP model is also less complete, as it does not account for the contribution of internal components such as the device electronics that are present in the irradiation line.

**Figure 13.** Validation measurements for the different detectors and irradiation configurations: System S1 and CRS1000 in the monoenergetic beams at PIAF, and CRS2000/B in the calibration sources at CIEMAT. Ratios are shown between the responses obtained from the calibrated MC model (corrected by the corresponding scaling factor  $\overline{S}$ ) and the experimental data.

290

300

305

#### 6 Conclusions

In this work, we conducted an experimental campaign to characterize commercial cosmic-ray neutron sensors (CRNS) using reference neutron fields based on radioactive sources such as <sup>252</sup>Cf, <sup>241</sup>Am-Be, and <sup>252</sup>Cf moderated with light and heavy water. These experiments were performed at two metrology institutes: PTB and CIEMAT.

The source-based reference fields provide a broad energy spectrum, enabling calibrations across the entire range relevant for CRNS applications. This campaign represents a first step toward the development of a metrological framework for CRNS calibration under controlled laboratory conditions. It also highlights current limitations, key requirements, and future steps necessary for establishing a robust calibration protocol.

The experimental detector responses were employed to validate the Monte Carlo (MCNP) models, providing insight into the detectors' behavior. The results show agreement between simulations and experimental data within 4–12 %, which is consistent with the estimated experimental uncertainties. A summary of the devices, simulation tools, and the obtained calibration factors is presented in Table 2.

| Manufacturer    | Device model | Simulation/Validation | Simulation code | Nuclear data  | MC adjustment factor $\overline{S}$ | $\operatorname{u}(\overline{S})$ |
|-----------------|--------------|-----------------------|-----------------|---------------|-------------------------------------|----------------------------------|
| Hydroinnova     | CRS1000      | PTB                   | MCNP6.1         | ENDF/B-VIII.0 | $0.79 \pm 0.07$                     | 0.09                             |
| Hydroinnova     | CRS2000/B    | CIEMAT                | MCNP6.2         | ENDF/B-VIII.0 | $1.34 \pm 0.05$                     | 0.04                             |
| StyX Neutronica | System S1    | PTB                   | MCNP6.1         | ENDF/B-VIII.0 | $1.26 \pm 0.16$                     | 0.12                             |

**Table 2.** Summary of the characterized detectors, including the simulation code and version used, the institution responsible for the simulation and validation, and the calibration factors obtained from the experimental data.

As a further validation, the corrected MCNP models were applied to simulate detector responses in PTB's monoenergetic neutron fields, with energies ranging from 24 keV to 5 MeV. While irradiations with isotopic sources provide a basis for deriving calibration factors, the monoenergetic fields allow pointwise validation of the simulated response function and assessment of the applied corrections.

It should be noted that these detectors, designed for outdoor deployment in environmental neutron fields, are optimized for epithermal and low-energy neutrons ( $

315

Considering these factors, there is a clear need for a dedicated reference neutron field specifically designed for CRNS calibration. Such a field should provide a broad energy spectrum, an appropriate counting rate, and homogeneity over large areas to accommodate the typical dimensions of these devices. Furthermore, given the detectors' sensitivity to scattered neutrons, suitable shadow cones or a well-characterized room-return component are required.

While the present measurements are subject to significant uncertainties, this campaign constitutes the first systematic validation of commercial CRNS detectors in laboratory-based neutron reference fields, marking an important step toward the development of dedicated calibration facilities that will allow higher precision in the future.

Data availability. The datasets that support the findings of this study are available from the corresponding author upon reasonable request.

320 Author contributions. MZ, RMV, PB, MK, MR, ZV, JW contributed to the conceptual design and development of the research project. MAMC, RMV, PB, ADC, MD, MK, BL, ZV, and MZ contributed to the preparation and execution of the experimental campaign, including data collection. MAMC and RMV carried out the analysis, evaluation, and interpretation of the data, with additional support and supervision from MR, BL, and MZ. Simulations were prepared by MAMC and RMV, with support from MK, JW, MR, BASA, PB, and ZV. MAMC wrote the original draft of the manuscript, while RMV, MK, BL, MD, MZ, and MR contributed to its revision. All authors took part in discussions and interpretation of the results.

Competing interests. The authors declare that they have no conflict of interest.

Acknowledgements. The authors thank Andre Lücke, Thorsten Klages, Sebastian Fässer and the operators of the PIAF facility for their dedication, support, and help during the experimental campaign.

The authors acknowledge the use of AI-assisted tools (ChatGPT, OpenAI) for language revision.

The SoMMet project has received funding from the European Partnership on Metrology, co-financed from the European Union's Horizon Europe Research and Innovation Programme and by the Participating States (funder name: European Partnership on Metrology, funder ID: 10.13039/100019599, grant 21GRD08 SoMMet).

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
