# Peer review of "Response characterization of Cosmic-Ray Neutron Sensors in neutron metrology reference fields"

_EGUsphere, 2025_

## Referee Comment (RC1)

**Response characterization of Cosmic-Ray Neutron Sensors in neutron metrology reference fields**

**Comments to authors**

General comments

I found this to be a clearly written and logically laid out manuscript. My specific comments are largely minor suggestions for alterations to make the text easier to understand for a non-expert in the field, and to avoid possible misunderstandings.

I do also have a small number of queries about exactly what is meant at some points. To explain these I have outlined below my understanding of what you have done.

In section 3 you describe calculations of the responses of three types of commercial CRNS instruments for plane parallel beams incident from various directions. This is a logical approach as this is the way instrument responses are defined. It is also the data required for linking neutron count rates with soil moisture via calculations of the spectrum and angular properties of the neutron field at the detector as a function of soil moisture.

The instrument response calculations need to be verified against experiment. Unfortunately monoenergetic parallel beams are not available so you perform measurements with radionuclide sources. For these fields there are two problems. The fields are not plane parallel beams, and there is the issue of the scatter component. As the usual scatter correction techniques outlined in ISO 8529 cannot be used you are forced to use for your calibration the field including scatter and allowance derived by calculating the full field characteristics (scatter and angular dependence). The results indicate that correction factors $S$ have to be introduced to get agreement between measurement and calculations, and these vary between 0.79 and 1.34. You say at line 215 that the differences from 1 "are mainly attributed to limitations in the completeness of the model geometries". You must therefore assume the uncertainties in the neutron field calculations are much smaller. Some justification of this assumption should be made. I also think you need to say a little more about these $S$ values. You presumably have more details about the detectors than you were allowed to publish. Do you have any ideas of the reasons for the discrepancies, wrong gas pressures, incorrect moderator thicknesses, etc? One objective of future work would presumably be to explain these $S$ values and get them closer to 1.

Having derived the $S$ values you perform measurements with monoenergetic beams at PTB. Because of the large low-scatter environment there you are able to make measurements, for the moderated detectors at least, that are close to what you would like to perform to validate the calculations. With the $S$ values derived previously you get good agreement. I assume that for both sets of calculations, those for the plane parallel beam and those for the source calibrations, the same geometrical model of the detectors was used. This is an important point and should be emphasised.

Specific comments

Lines 4-5: I don't think your sentence, "Despite its importance, existing monitoring methods operate at different spatial and temporal scales and remain poorly harmonized, leading to inconsistencies among datasets" actually says what you mean. I would suggest something like: 'Existing monitoring methods operate at different spatial and temporal scales, and despite the importance of soil moisture data, they remain poorly harmonized, leading to inconsistencies among datasets.

L 7: something seems to have gone wrong on this line.

L 15: I didn't think CIEMAT had monoenergetic neutron reference fields in the 24 keV to 5 MeV range.

L 25: is there a web address for the reference WMO, UNEP, ISC, IOC-UNESCO, 2022?

L 30: why "satellite products"? Seems an odd phrase. Do you just mean 'satellite observations'?

L 43: Not sure the word "environmental" helps here. It suggests neutrons other than cosmic ray induced ones.

Ls 46-47: You note that environmental conditions have an impact on measurements. You do not cover these effects in the paper. It may be worth saying this, and maybe even that they will be the subject of further study if it is the case.

L 40: What does "Visited on January 2025" mean here?

L 73: Fig. 7 is out of order in the text. Alos, Fig. 7 is not ideal. The ceiling lighting draws the attention making it difficult not familiar with the facility to pick out the important features, i.e. the detector and $D_2O$ moderated source. I guess there is probably nothing that can be done about this at this stage.

L 78: where you say the "unit is 154 cm tall". However, the unit is mounted horizontally in Fig. 7 so '154 cm long' might be more appropriate.

L 113: the external dimensions of the CRS 1000 are not the same here as at line 73.

L 114: "internal scattering" needs clarification. I assume it means neutrons scattered within the containment box of the detector are included.

Fig. 1: in the figure NH connectors are indicated. I assume these are particular types of detectors, but this information is not important in the context of this paper. The labelling would be more informative if it just said 'signal and HV connectors'.

In reading about the CRS 1000 it occurred to me that the way the instrument is mounted in the field is not clearly stated for the various devices.

Fig. 5: one feature is labelled as "Cu (inner coating)". The feature looks to be much thicker than a coating. Is in not the wall of the gas detector?

L 151: I don't think Q is a universally accepted symbol for source emission rate. If you want to use it, it should be defined here as it is at line 182.

Fig. 7 caption: you probably need to clarify that the cadmium layer only covers one hemisphere of the $D_2O$ moderator.

L 185: "vertical irradiations" is not an ideal description as the tube is mounted horizontally in Fig. 9.

L 246: was it a specially constructed shadow cone?

Conclusions: It might be worth saying that future work should concentrate on finding reasons why $S$ is not 1. At line 295 you say, "The experimental detector responses were employed to validate the Monte Carlo (MCNP) models". To me this phrase encapsulates the core of what has been done. The validity of the soil moisture measurements depends critically on the calculations for the response of the detectors used, and the more that can be done to validate these calculations the better.